# Cytochrome *c* Oxidase Subunit 4 Isoform Exchange Results in Modulation of Oxygen Affinity

**DOI:** 10.3390/cells9020443

**Published:** 2020-02-14

**Authors:** David Pajuelo Reguera, Kristýna Čunátová, Marek Vrbacký, Alena Pecinová, Josef Houštěk, Tomáš Mráček, Petr Pecina

**Affiliations:** 1Department of Bioenergetics, Institute of Physiology, Czech Academy of Sciences, 14220 Prague 4, Czech Republic; d.pajuel@gmail.com (D.P.R.); kristyna.cunatova@fgu.cas.cz (K.Č.); marek.vrbacky@fgu.cas.cz (M.V.); alena.pecinova@fgu.cas.cz (A.P.); josef.houstek@fgu.cas.cz (J.H.); 2Department of Cell Biology, Faculty of Science, Charles University, 12000 Prague 2, Czech Republic

**Keywords:** mitochondria, OXPHOS, respiratory chain, cytochrome *c* oxidase, COX, COX4 isoforms, COX4i2, oxygen affinity, p_50_, oxygen sensing

## Abstract

Cytochrome *c* oxidase (COX) is regulated through tissue-, development- or environment-controlled expression of subunit isoforms. The COX4 subunit is thought to optimize respiratory chain function according to oxygen-controlled expression of its isoforms COX4i1 and COX4i2. However, biochemical mechanisms of regulation by the two variants are only partly understood. We created an HEK293-based knock-out cellular model devoid of both isoforms (COX4i1/2 KO). Subsequent knock-in of COX4i1 or COX4i2 generated cells with exclusive expression of respective isoform. Both isoforms complemented the respiratory defect of COX4i1/2 KO. The content, composition, and incorporation of COX into supercomplexes were comparable in COX4i1- and COX4i2-expressing cells. Also, COX activity, cytochrome *c* affinity, and respiratory rates were undistinguishable in cells expressing either isoform. Analysis of energy metabolism and the redox state in intact cells uncovered modestly increased preference for mitochondrial ATP production, consistent with the increased NADH pool oxidation and lower ROS in COX4i2-expressing cells in normoxia. Most remarkable changes were uncovered in COX oxygen kinetics. The p_50_ (partial pressure of oxygen at half-maximal respiration) was increased twofold in COX4i2 versus COX4i1 cells, indicating decreased oxygen affinity of the COX4i2-containing enzyme. Our finding supports the key role of the COX4i2-containing enzyme in hypoxia-sensing pathways of energy metabolism.

## 1. Introduction

Cytochrome *c* oxidase, the terminal enzyme of the electron transport chain, is an indispensable part of mitochondrial machinery needed for ATP production in mammalian cells (OXPHOS). In addition to three mitochondria-encoded subunits, which are necessary for COX catalytic function, eleven nuclear-encoded subunits build up the COX enzyme and participate in the regulation of COX enzyme activity, as well as in the regulation of the whole OXPHOS system [1]. The crucial position in the respiratory chain pathway, a large drop of Gibbs free energy during enzyme turnover, and relatively low in vivo reserve capacity [2] predispose COX to serve as a mitochondrial OXPHOS regulator. This is indeed reflected by the emergence of numerous regulatory subunits during the evolution of individual eukaryotic lineages [3], as well as by the discovery of numerous post-translational modifications signifying that COX has become a target of signaling pathways [4]. In mammals, six COX subunits have isoforms with expression regulated in developmental, tissue-specific (COX6a, 7a, 8, and 6b), and environmental manners (COX4, NDUFA4). Subunits 6a, 7a, and 8 all exist in two variants, the L (liver) isoform is expressed ubiquitously, and the H (heart) isoform is expressed postnatally in heart and skeletal muscle [5]. In contrast to the L isoforms, the triplet of H isoforms functions as an ATP/ADP sensor and modulates COX turnover and its H^+^/e^−^ efficiency [5]. An additional 7a variant, COX7a2L, is responsible for the association of complexes III and IV into supercomplexes [6]. A second isoform of subunit 6b was discovered in mammalian testis (6b-2). Subunit 6b forms part of the cytochrome *c* binding site, so the testis isoform was hypothesized to evolve in association with the testis isoform of cytochrome *c* [7].

The largest of the nuclear encoded subunits, COX4, is ideally predisposed to serve as a regulatory factor thanks to its interactions with other subunits throughout multiple parts of the COX enzyme. Its C-terminal part protrudes into the intermembrane space where it interacts with COX2 and helps shape the docking site for cytochrome c. The COX4 transmembrane helix is tightly associated with catalytic subunit COX1 and runs parallel alongside helices of nuclear-encoded subunits COX7b, COX8a, and COX7c. Moreover, the large membrane extrinsic domain of COX4 is located in the mitochondrial matrix and may function as a metabolic sensor. Indeed, COX4 has long been recognized as a target of allosteric ATP binding that decreases enzyme turnover, and this mechanism has been denoted as a second mechanism of respiratory control [8]. Furthermore, it was reported that ATP binding is abolished upon phosphorylation of serine 58 in COX4-1 by the protein kinase A pathway, with all components located in the mitochondrial matrix, and is positively regulated by CO_2_ and thus may serve to match the rates of the Krebs cycle with the respiratory chain [9]. In yeast, two isoforms of the COX4 homolog COXV were discovered (COXVa and COXVb). Their ratio is exclusively regulated by oxygen concentration, with COXVb being expressed under hypoxia [10]. The second isoform of mammalian COX4, COX4i2, discovered in 2001, was found to be predominantly expressed in the lung, and to a minor extent in the heart and brain [11]. COX4i1 and COX4i2 are highly homologous in the C-terminal region; however, significant differences exist in the N-terminal part comprising the matrix domain. COX4i2 lacks the regulatory S58 residue, and in contrast to COX4i1, contains three cysteine residues that may form disulphide bridges under oxidative conditions and thus function as redox sensors [11]. Importantly, and similarly to yeast, the mammalian COX4 isoform expression is regulated by oxygen availability [12,13,14]. Interestingly, the gene duplication event giving rise to the isoforms occurred independently in both eukaryotic lineages. This suggests that the COX4 position in the enzyme enables the isoform switch to serve as a homeostatic response to optimize respiratory chain function according to oxygen availability. In support of that, studies on the mouse COX4i2 knockout model revealed that this protein is essential for acute hypoxic pulmonary vasoconstriction [15,16]. Later it was shown that COX4i2 is also likely involved in oxygen sensing by carotid bodies [17,18]. In human cells, the function of COX4 isoforms has been partially characterized in HeLa and HEK293 cells where COX4i2 expression under hypoxia increased enzyme activity and decreased ROS production from mitochondria [13]. Increased activity of the COX4i2-containing enzyme was also found in astrocytes, associated with decreased sensitivity to allosteric inhibition by ATP [19]. In contrast, a recent study in glioblastoma cells reported increased activity of the COX containing COX4i1 isoform [20]. Some of these contradictory findings may be attributed to cellular models used in these studies, which were largely based on shRNA knock-down technology. Such an approach cannot completely abolish the expression of targeted proteins, and therefore, at least a minor expression of both isoforms is always present.

To unambiguously characterize the functional differences between COX4 isoforms, we constructed cellular models with exclusive knock-in of respective isoforms in the background of double CRISPR-Cas9-mediated COX4i1/2 knockout in HEK293 cells. We report that COX4 isoforms 1 and 2 are able to modulate COX oxygen affinity and can affect mitochondrial OXPHOS and the redox state even under normoxic conditions.

## 2. Materials and Methods

### 2.1. Cell Culture

HEK293 (human embryonic kidney) cells (ATCC® CRL-1573™) were cultivated under standard conditions (37 °C, 5% CO_2_ atmosphere) in nutrient-rich DMEM/F-12 (31331-028, Thermo Fischer Scientific, Waltham, MA, USA) supplemented with 10% (*v*/*v*) FBS (Thermo Fischer Scientific, 10270-106), 40 mM HEPES, 50 µM uridine, and antibiotics (100 U/mL penicillin, 100 μg/mL streptomycin, Thermo Fischer Scientific, 15140-122) to facilitate growth of highly glycolytic knock-out cells.

### 2.2. Preparation of HEK293 Cellular Knock-Out Models

Double COX4i1 and COX4i2 gene knock-out (KO) was introduced into HEK293 cells by CRISPR (Clustered Regularly Interspaced Short Palindromic Repeats) knock-out technology employing Cas9-D10A paired nickase with two chimeric RNA duplexes [21]. First, the COX4i2 KO HEK293 model was established and was subsequently used to introduce COX4i1 KO to obtain COX4i1/2 KO cells.

gRNAs for COX4i1 and COX4i2 obtained from a commercially available library (Merck, Darmstadt, Germany) were used for knock-out preparation. The HSR0002687616/HSL0002687619 gRNA pair was chosen to generate COX4i2 KO and the HSL0001377215/ HSR0001377224 gRNA pair for COX4i1 KO. gRNAs were received in the form of the expression plasmid (U6-gRNA, expressed by the U6 promoter), along with the expression plasmid for Cas9-D10A nickase fused with GFP (Merck). The efficiency of different combinations of gRNA pairs for the Cas9-D10A nickase system was determined by SURVEYOR assay (Integrated DNA Technologies, Coralville, IA, USA). Cells were transfected with relevant plasmids using Metafectene Pro (Biontex Laboratories GmbH, München, Germany), incubated for 72 h, trypsinized, and then diluted to a concentration of 20 cells/mL. This suspension was aliquoted (100 µL) into 96-well plates. Wells containing single-cell colonies were identified and further cultivated. Confluent cells in 6-well plates were harvested and used for selection of a knock-out clone by sequencing of CRISPR-targeted sites to identify nonsense mutations resulting in premature stop codons and by SDS-PAGE WB to confirm the complete absence of COX4 isoforms at the protein level.

### 2.3. COX4 Isoform Overexpression

COX4i1 (CCDS10955.1) and COX4i2 (CCDS13187.1) isoforms of the human COX4 subunit full-length cDNAs including the 3′ end sequence encoding the FLAG tag (DYKDDDDK) were obtained as synthetized DNA strands with 5′ and 3′ overhangs containing restriction sites used for insertion into the pcDNA3.1^+^ mammalian expression vector (Thermo Fischer Scientific). COX4i1-FLAG or COX4i2-FLAG constructs were transfected into COX4i1/2 KO cells using Metafectene Pro (Biontex Laboratories GmbH); the selection of stably-transfected cells was started 48 h post-transfection with 1 mg/mLG418. Single-cell clones were later established from G418-resistant cells, and three clones of each COX4i1 and COX4i2 knock-in cells were used for the following experiments.

### 2.4. COX4i2 Site-Directed Mutagenesis

The COX4i2-FLAG pcDNA3.1^+^ construct was used to introduce mutations leading to single amino acid (AA) substitutions of three individual cysteine residues at positions 40, 54, and 108. Each cysteine was replaced by three AA variants, serine, representing a sterically similar amino acid where only sulphur is replaced by oxygen, or glycine and valine, which represent amino acids present at homologous positions in COX4i1. Mutations were introduced using the QuikChange Lightning kit (Agilent, Santa Clara, CA, USA), using primers suggested by the manufacturer’s on-line primer design tool. Mutagenesis was confirmed by sequencing, and mutated constructs were used for knock-in into COX4i1/2 KO cells as was the wild-type COX4i2-FLAG vector.

### 2.5. Electrophoretic Analyses

Sodium dodecyl sulphate polyacrylamide gel electrophoresis (SDS-PAGE), specifically tricine SDS-PAGE, was used for separation of proteins according to their molecular weight under denaturing conditions [22]. Frozen aliquots of harvested cellular pellets were suspended in SLB buffer (sample lysis buffer; 2% (*v*/*v*) 2-mercaptoethanol, 4% (*w*/*v*) SDS, 50 mM Tris (pH 7,0), 10% (*v*/*v*) glycerol, 0,02% Coomassie Brilliant Blue R-250) to a final concentration of 2–5 mg protein/mL. Samples were sonicated and then incubated for 20 min at 40 °C. Samples (10–50 µg of protein as indicated) were separated on 12% gel using a Mini-PROTEAN III apparatus (Bio-Rad, Hercules, CA, USA).

Blue-native (BN-PAGE) electrophoresis [23] was used for analysis of native protein complexes. For protein complex solubilization, mild non-ionic detergent digitonin was used, as it maintains OXPHOS complexes and their supercomplexes and enables their detection.

Mitochondria from HEK293 cells were released by hypotonic shock and isolated using differential centrifugation as described [24] and then stored as dry pellets at −80° C. Mitochondrial pellets were resuspended in solubilization buffer (50 mM NaCl, 50 mM imidazole/HCl, 2 mM 6-aminocaproic acid, 1 mM EDTA; pH 7,0) at a protein concentration 10 mg/mL. Digitonin (20% (*w*/*v*) stock solution) was added at 6 g/g protein, and solubilization proceeded on ice for 10 min. Samples were centrifuged at 30,000× *g* (20 min, 4 °C), and supernatant containing released membrane proteins was collected. Then, glycerol was added to a final 5% (*v*/*v*) concentration, and Coomassie Brilliant Blue G-250 (CBB) was supplemented according to a detergent/dye ratio of 8:1 (*w*/*w*). Samples were separated on 5–16% gradient polyacrylamide gel using a Mini-PROTEAN III apparatus (Bio-Rad).

### 2.6. Western Blot and Immunodetection

Proteins separated by electrophoresis were transferred onto PVDF (polyvinylidene difluoride) membrane (Immobilon FL 0.45 µm, Merck) by semi-dry electroblotting using a Transblot SD apparatus (Bio-Rad). After blotting, the PVDF membrane was washed for 5 min in TBS (150 mM Tris-HCl, 10 mM NaCl; pH 7,5) and blocked in 5% (*w*/*v*) fat-free dry milk diluted in TBS for 1 h. Then, the membrane was washed 2 × 10 min in TBST (TBS with 0,1% (*v*/*v*) detergent Tween-20). For immunodetection, the membrane was incubated for 2 h in primary antibody (diluted in TBST) at room temperature. The following primary antibodies were used in the study: FLAG (Merck F1804), complex IV (COX1: Abcam 14705, Cambridge, UK, COX2: Abcam 110258, COX4i1: Abcam 14744, COX4i2: Abnova H00084701-M01, Taipei City, Taiwan, COX5a: Abcam 110262, COX6c: Abcam 110267), complex I (NDUFA9: Abcam, 14713), complex II (SDHA: Abcam 14715), complex III (Core 2: Abcam 14745), and citrate synthase (Abcam 129095). For quantitative detection, the corresponding infra-red fluorescent secondary antibodies (Alexa Fluor 680, Thermo Fisher Scientific; IRDye 800, LI-COR Biosciences, Lincoln, NE, USA) were used. Detection was performed using the fluorescence scanner Odyssey (LI-COR Biosciences) and signals were quantified by ImageLab software (Bio-Rad).

### 2.7. Immunoprecipitation and Mass Spectrometry Analysis

Whole-cell digitonin solubilizates (2 g/g of protein) from COX4i1-FLAG KI, COX4i2-FLAG KI, and COX4i1/2 KO (background control not containing FLAG-tagged protein) cells were immunoprecipitated with anti-FLAG M2 magnetic beads (Merck M8823). Washed beads with captured immunocomplexes were digested “on beads” with trypsin using the sodium deoxycholate procedure as described in [25]. Desalted peptide digests were separated on a 50 cm C18 column using a 1 h elution gradient and were analyzed in an Orbitrap Fusion (Thermo Fisher Scientific) mass spectrometer. Resulting raw files were processed in MaxQuant (v. 1.6.6.0, maxquant.org) with the label-free relative quantification (LFQ) algorithm MaxLFQ [26]. Downstream analysis and visualization were performed in Perseus (maxquant.org/perseus/) [27].

### 2.8. Cytochrome c Oxidase Activity

COX activity was analyzed using Oxygraph-2k (Oroboros, Innsbruck, Austria) in mitochondrial extracts essentially as described [28]. Mitochondria isolated from cells were suspended in buffer composed of 10 mM K-HEPES (pH 7.4), 40 mM KCl, 2 mM EGTA, 1% Tween 20, and protease inhibitor cocktail (Merck), sonicated, and centrifuged at 30,000× *g* for 2 min. Then, 200 µg protein of supernatants was added to 2 mL oxygraph chambers. COX activity was followed as the rate of oxygen consumption in the presence of 20 mM ascorbate serving as a reductant for cytochrome *c* (bovine, Merck) titrated sequentially to reach final concentrations of 2.5–30 µM. Parallel measurement with COX activity fully inhibited by 0.5 mM KCN was always performed to subtract non-specific oxygen consumption due to autoxidation of ascorbate. A hyperbolic Michaelis-Menten function was fitted to experimental data using Prism 8 software (GraphPad, San Diego, CA, USA).

### 2.9. High-Resolution Respirometry

Mitochondrial respiration was measured at 30 °C as described in [29] using Oxygraph-2k (Oroboros, Innsbruck, Austria). Freshly harvested cells (0.6 mg protein) were suspended in 2 mL of KCl medium (80 mM KCl, 3 mM MgCl_2_, 1 mM EDTA, 5 mM K-Pi, 10 mM Tris-HCl; pH 7.4), and digitonin (0.05 g/g) protein was used to permeabilize the plasma membrane. For measurements, the following substrates and inhibitors were used: 2 mM malate, 10 mM pyruvate, 10 mM glutamate, 10 mM succinate, 10 mM glycerol 3-phosphate, 1 mM ADP, 0.5 µM oligomycin, 100–300 nM FCCP, 0.5 µM rotenone, 10 mM malonate, 0.25 µM antimycin A, 2 mM ascorbate, 1 mM TMPD, 0.5 mM KCN. The oxygen consumption was expressed in pmol O_2_/s/mg protein.

Affinity to oxygen was determined by the p_50_ value, the partial oxygen pressure (pO_2_), at which the cellular respiratory rate is half-maximal, as described before [30]. Briefly, the volume-specific rate of oxygen consumption (oxygen flux) was calculated as the negative slope of the oxygen concentration recorded at 1 s time intervals. The signal was deconvoluted with the exponential time constant of the oxygen sensor (3 to 5 s) and corrected for the instrumental background, which is a linear function of experimental pO_2_ and results from oxygen consumption by the sensor and oxygen back-diffusion from low-capacity oxygen reservoirs. The p_50_ parameter was obtained from a hyperbolic function, J_02_ = (J_MAX_. pO_2_)/(p_50_ + pO_2_), fitted over the low oxygen range of 0 to 1.1 kPa. All calculations were performed using routine functions of Datlab 2 software (Oroboros) [31].

### 2.10. XF Seahorse Bioenergetics Analyser Measurement

Parallel measurement of mitochondrial respiration and the glycolytic rate was performed using a Seahorse XFe24 analyzer (Agilent). One day prior to measurement, 3 × 10^4^ cells were seeded in poly-L-lysine coated wells of the measuring plate and were cultivated in DMEM/F12 medium (see above). The next day, the microplate was washed with 1 mL of XF assay medium (modified DMEM, pH 7.4, 37 °C); 500 μL of the same medium with 0.2 % (*w*/*w*) BSA was pipetted, and the microplate was incubated at 37 °C for 30 min. Meanwhile, an XFe24 sensor cartridge was prepared by injection of substrates and inhibitors according to the described protocol [32] to record metabolic rates with endogenous substrates before any addition (basal), and after subsequent additions reaching final concentrations of 10 mM glucose (Glu), 1 µM oligomycin (Oligo), 1 µM FCCP (FCCP), and inhibitor cocktail of 1 µM rotenone, 1 µg/mL antimycin A, and 100 mM 2-deoxyglucose (Rot+AA+2DG). For precise normalization of rates according to cell counts, immediately after the Seahorse run, cell nuclei were stained by Hoechst 33342 (final concentration 5 µg/mL). Images of whole wells were acquired by a Cytation 3 Cell Imaging Reader (BioTek, Winooski, VT, USA) and were analyzed using Gen5 software (BioTek) to obtain cell counts for each well.

### 2.11. ROS Production

ROS production was estimated as 1 μM CM-H_2_DCFDA (chloromethyl derivative of 2′,7′-dichlorodihydrofluorescein diacetate, Thermo Fisher Scientific) fluorescence as in [33]. Cells (3.x10^5^) were seeded in poly-L-lysine coated 24-well plates, and the next day, cells were incubated for 2 h at 37 °C with CM-H_2_DCFDA. The increase in the fluorescence was recorded with an Infinite M200 plate reader (Tecan Group Ltd., Männedorf, Switzerland) at 495/525 nm.

### 2.12. NAD^+^/NADH Ratio

Redox balance in terms of NAD^+^/NADH was measured in cell lines using the NAD^+^/NADH-Glo assay (Promega) essentially as described [34]. Briefly, 5x10^3^ cells were seeded into 96-well plates one day prior to measurement. NAD^+^ and NADH concentrations in cell extracts were determined according to the manufacturer’s protocol, and the luminescence signal was recorded with an Infinite M200 plate reader (Tecan Group Ltd.).

## 3. Results

### 3.1. Comparable COX Content and Composition in COX4i1 and COX4i2 KI Cells

To obtain an optimal research model for unbiased comparison of functional features of COX isozymes distinguished solely by the presence of either COX4i1 or COX4i2, we constructed HEK293-based cell lines with CRISPR/Cas9 mediated knock-out of both COX4 isoforms (COX4i1/2 KO), followed by stable knock-in of either isoform. Recombinant COX4 isoforms were expressed with the C-terminal FLAG tag to facilitate isoform content comparison as well as their immunoprecipitation using the same antibody.

For this study, three representative single-cell clones of both COX4i1 and COX4i2 knock-in cell lines (4i1 KI and 4i2 KI, respectively) were chosen. Exclusive expression of respective COX4 isoforms was verified in whole cell lysates by SDS-PAGE WB followed by immunodetection (Figure 1A). Of note here is that COX4i2 was not detected in parental HEK293 cells, suggesting that under normoxic conditions, HEK293 cells express exclusively the COX4i1 isoform. The content of the mtDNA-encoded subunit COX2 was identical in 4i1 and 4i2 KI, in both cases reaching approximately 50% of the COX2 level in parental HEK293 cells when normalized to the citrate synthase content (Figure 1A,C). Similarly, levels of other tested COX subunits, COX5a and COX6c, were comparable in 4i1 and 4i2 KI cell lines (Figure 1B,C). These data indicate a similar COX content in 4i1 and 4i2 KI, even though the quantification of the FLAG tag signal present on recombinant COX4 isoforms showed an approximately 2.5-fold higher COX4i2 steady-state level compared to COX4i1 in the respective KI cell lines (Figure 1B,D). To compare the content of other OXPHOS complexes in whole cell lysates, SDS-PAGE WB immunodetection was also performed with representative subunits of complexes I (NDUFA9), II (SDHA), and III (core 2). Their amount was not significantly different in 4i1 and 4i2 KI, corresponding to 85, 65, and 80% of the control levels of complexes I, II, and III, respectively (Figure 1B,C).

To analyze the assembly status of COX complexes and their interactions in OXPHOS supercomplexes, isolated mitochondria from 4i1 and 4i2 KI cells were solubilized with the mild detergent digitonin and were analyzed by BN-PAGE. Immunodetection of COX1, the core catalytic subunit, revealed that the COX content and its distribution between enzyme monomeric and dimeric forms, as well as its association into respiratory supercomplexes is identical for 4i1 and 4i2 KI (Figure 1E). In accordance with the quantification of the COX subunit contents (Figure 1C), the amount of the native COX complexes in the 4i1 and 4i2 KI cell lines corresponded to 54% and 61% of the COX level in parental HEK293, respectively.

Since crystal structure data are only available for the enzyme containing the COX4i1 isoform, we further focused on detailed characterization of native COX complexes containing either COX4i1 or COX4i2 isoforms. This was performed by mass spectrometric (MS) analysis of proteins immunoprecipitated from whole cell digitonin solubilizates using antibody directed against the FLAG tag present in recombinant COX4 isoforms. Comparison of COX4 interacting proteins between 4i1 and 4i2 KI revealed that the majority of COX subunits as well as COX assembly factors was associated with both COX isozymes (Figure 1F). Except for obvious hits of COX4i1 and COX4i2 in their respective KI cell lines, only two other mitochondrial proteins showed a significantly different association with either of the COX isozymes enriched in the COX4i1 KI cell line—assembly factor COX11 and m-AAA protease AFG3L2. These proteins associate with COX complexes during enzyme biogenesis to either facilitate copper insertion into COX1 in the case of COX11 or to impose protein quality control in the case of AFG3L2, and both have not been previously reported as components of mature COX complexes. Therefore, with the exception of COX4 isoforms, the fully assembled COX in both 4i1 and 4i2 KI cell lines had identical composition. In fact, increased association of COX4i1 with AFG3L2 protease could explain the differential content of the FLAG-tagged isoforms. While unassembled isoform 1 was efficiently degraded, a portion of isoform 2 accumulated even outside the COX complex.

In summary, electrophoretic and MS analyses demonstrated that cell lines with exclusive expression of either COX4i1 or COX4i2 isoforms of the COX4 subunit had a comparable enzyme content and composition, and thus represent a convenient model to study functional differences conferred by these isoforms.

### 3.2. Identical Cytochrome c Oxidase Activity in KI Cell Lines

Respirometric assay of COX activity during titration of increasing concentrations of cytochrome *c* in mitochondrial solubilized extracts was employed to obtain maximal activity rates as well as data on the cytochrome *c* binding kinetics of the COX complex in 4i1 and 4i2 KI cells.

In both knock-in cell lines, COX activity displayed a hyperbolic response to increasing substrate concentrations. In terms of maximal activity rates at 30 µM concentration of cytochrome *c*, 4i1 and 4i2 KI cell lines displayed equal values, corresponding to 55% and 52% of control HEK293 values, respectively (Figure 2A). When the data were normalized to the content of the COX2 subunit in the solubilized extracts, maximal rates in KI cell lines matched the controls (Figure 2B). These findings indicate that specific activities of cytochrome *c* oxidation of COX4i1- and COX4i2-containing COX isozymes in KI cell lines cannot be distinguished from each other and are perfectly analogous to the activity of the wild-type enzyme in parental HEK293 cells. Furthermore, COX affinity to cytochrome *c* was assessed by fitting a hyperbolic Michaelis–Menten function through experimental data (Figure 2B). The obtained K_m_ values of 9.9 ± 2.26 and 9.0 ± 2.39 µM for 4i1 and 4i2 KI cells, respectively, were not significantly different. Overall, the isoform switch between COX4i1 and COX4i2 isoforms did not result in alterations of affinity and maximal velocity of cytochrome *c* oxidizing activity of the COX complex.

### 3.3. Decreased COX Oxygen Affinity in COX4i2 KI cells

To analyze the COX function in the context of a fully operating respiratory chain, mitochondrial respiration was measured in situ in cells permeabilised with digitonin. The complex substrate/inhibitor titration protocol (Figure 3A), including parallel electron input to both major dehydrogenases in the HEK293 respiratory chain, complexes I and II, as well as to glycerol-3-phosphate dehydrogenase, was employed to obtain the capacities of oxidative phosphorylation (OXPHOS, i.e., coupled mitochondrial respiration in the presence of ADP and saturating substrate concentrations), electron-transport chain capacity (ETC, i.e., mitochondrial respiration uncoupled by FCCP under saturating substrate concentrations), and COX capacity (COX, i.e., oxygen consumption with artificial substrates ascorbate and TMPD donating electrons to cytochrome *c*). Furthermore, two aerobic/anoxic transitions were included, in OXPHOS and ETC states, to obtain data for analysis of oxygen kinetics.

In the OXPHOS state, 4i1 and 4i2 KI cells displayed comparable respiratory rates, 140.9 ± 7.04 and 140.7 ± 10.22 pmol O_2_/s/mg protein, respectively, that were not significantly different from the OXPHOS capacity in control HEK293 cells (161.4 ± 18.65 pmol O_2_/s/mg protein) (Figure 3B). Likewise, 4i1 and 4i2 KI and HEK293 cells could not be distinguished by respiratory rates in the uncoupled state (ETC, 284.4 ± 16.11, 270.8 ± 23.72, and 327.3 ± 51.64 pmol O_2_/s/mg protein, respectively) (Figure 3B). These data indicate that selective knock-in of either isoform into COX4 dKO cells fully complemented the capacities of electron transfer in the respiratory chain and, importantly, the wild-type level of mitochondrial ATP production of HEK293 cells. The lower COX content in KI cells relative to the control HEK293 was only manifested during analysis of the COX capacity with artificial substrates. While the COX capacity rates in 4i1 and 4i2 KI cells were not significantly increased above the value of the respective ETC capacities, a 1.5-fold increase in parental HEK293 cells was indicative of the COX reserve capacity (Figure 3B).

Oxygen kinetics were analyzed in OXPHOS and ETC states in terms of the p_50_ parameter, the partial pressure of oxygen at the half-maximal respiratory rate, thus analogous to K_M_ from classical enzyme kinetics. The p_50_ values in 4i1 KI cells, 0.0637 ± 0.01291 and 0.0486 ± 0.01436 kPa in OXPHOS and ETC states, respectively, were not significantly changed compared to control HEK293 cells, which do indeed contain the 4i1 isoform (Figure 3C). In contrast, p_50_ in 4i2 KI cells in the OXPHOS state (0.12015 ± 0.01793 kPa) and ETC states (0.0705 ± 0.01215 kPa) was increased 2-fold compared to 4i1 KI and parental HEK293 cells (Figure 3C). These data represent strong indication that the COX4i2 isoform confers lower oxygen affinity of the COX complex, thus decreasing the respiratory rate under limiting oxygen supply. Simulated values of the relative respiratory rate in the oxygen-sensitive range for 4i1 and 4i2 KI cells are shown in Figure 3D.

The most remarkable structural difference between isoforms 1 and 2 of COX4 may be the exclusive occurrence of three cysteine residues in COX4i2 (C48, C54, C108). To test whether the observed changes in oxygen affinity could be assigned to the presence or modification of individual cysteine residues, site-directed mutagenesis was employed to prepare three variants, glycine, valine or serine, of each of the cysteines (C48G, C48V, C48S, C54G, C54V, C54S, C108G, C108V, C108S). These mutant KI cells displayed varying ability to complement the content of COX (Figure 4A) and mitochondrial respiration (Figure 4B). In general, serine substitutions at each of three cysteine positions were able to maintain COX levels and respiratory rates comparable to knock-in with WT COX4i2. Nevertheless, the oxygen affinity did not respond to the site-specific substitutions, as C48S, C54S, and C108S KI cell lines displayed unchanged values of p_50_ compared to 4i2 KI (Figure 4C). These findings indicate that diminished oxygen affinity of COX4i2-containing oxidase is not due to intra- or intermolecular crosslink or posttranslational modification involving either of its three cysteine residues.

### 3.4. Energy Metabolism, ROS Production, and Redox Status in Intact Cells

To compare intact cell energy metabolism in 4i1 and 4i2 KI cell lines, mitochondrial respiration (oxygen consumption rate, OCR, Figure 5A) and glycolytic activity (extracellular acidification rate, ECAR, Figure 5B) were analyzed simultaneously using a Seahorse XFe24 Bioenergetic Analyzer. Metabolic rates were monitored in cells catabolizing only endogenous substrates in the beginning, followed by subsequent additions of 10 mM glucose and 1 µM oligomycin to inhibit mitochondrial ATP synthase, 1 µM FCCP to uncouple oxidative phosphorylation and stimulate maximal electron flux of the respiratory chain, and a combination of rotenone (1 µM), antimycin A (1 µM), and 2-deoxyglucose (100 mM) to inhibit both the respiratory chain and glycolytic pathway.

4i1 and 4i2 KI cells showed comparable values of basal OCR with glucose as a substrate (Figure 5A). The residual leak-driven respiration (oligomycin state) was similar in both KI cell lines, as well as the respiratory capacity assessed after FCCP addition (Figure 5A). Basal ECAR with glucose as a substrate showed a 1.5-fold higher value in 4i1 KI, but glycolytic capacity tested after oligomycin addition and the glycolytic reserve capacity were similar between 4i1 and 4i2 KI (Figure 5B). Finally, the OCR/ECAR ratio was calculated from rate values in the glucose state to assess cell preference for either glycolytic or mitochondrial ATP provision. This parameter was 1.4-fold higher in COX4i2 KI cells, indicating increased preference for ATP production by OXPHOS compared to COX4i1 KI cell lines (Figure 5C).

The mitochondrial respiratory chain is one of the major cellular producers of reactive oxygen species (ROS). To analyze whether the switch from COX4i1 to COX4i2 was associated with changes in oxidative stress, mitochondrial ROS production was measured in intact cells using the fluorescent probe H_2_-DCFDA that is sensitive to broad-spectrum ROS. Basal ROS production in intact 4i1 and 4i2 KI cell lines was compared with parental HEK293 cells and the COX4 dKO cell line. While 4i1 KI displayed similar values as HEK293 cells, 4i2 KI cells showed a 1.5-fold decreased rate of ROS production. (Figure 6A). Upon inhibition of complex III by antimycin A, which leads to reduction of the mitochondrial NADH pool, rates of ROS production increased approximately twofold (Figure 6A), but no significant differences were detected between the cell lines. These data indicate that the alterations recorded under basal conditions result from the current metabolic and redox status of mitochondria rather than increased disposition for ROS production in particular cell lines. Indeed, the ROS production rate difference between 4i1 and 4i2 KI cells correlated with values of the NAD^+^/NADH ratio, which was increased by 20% in 4i2 KI, indicating a less reduced NADH pool (Figure 6B). As expected, NAD^+^/NADH was extremely low in COX4i1/2 KO cells due to an inactive respiratory chain (Figure 6B). Unfavorable redox status correlated with increased ROS production in knockout cells (Figure 6A).

In summary, despite performing the studies under normoxic conditions, analysis of energy metabolism and the redox state in intact cells uncovered modestly increased preference for mitochondrial ATP production, consistent with increased NADH pool oxidation and lower ROS in 4i2 KI compared to 4i1 KI cells.

## 4. Discussion

Energetic metabolism of mammalian organisms is vitally dependent on aerobic respiration, with 80–90% of inhaled oxygen being utilized by cytochrome *c* oxidase, the terminal enzyme of the mitochondrial respiratory chain. The apparent K_m_ for oxygen ranges between 0.01 and 0.1 kPa, meaning that mitochondrial respiration is usually not limited by oxygen availability [35]. However, even with sophisticated oxygen delivery by the cardiovascular system, local pO_2_ can drop to levels as low as 0.3 kPa in some tissues. Therefore, oxygen reduction to water by cytochrome *c* oxidase occurs under conditions of oxygen availability varying by two orders of magnitude [36], which under extreme cases may result in incomplete COX saturation. Oxygen affinity of COX can be further diminished by allosteric factors such as NO [37], as part of mechanisms that can redirect oxygen utilization from respiration to other oxygen-handling enzymes or non-enzymatic reactions. The significance of oxygen availability for COX function was strongly supported by the discovery of oxygen-regulated expression of COX4i2, the second isoform of regulatory, nuclear-encoded subunit COX4. Oxygen regulation of COX4i2 expression is exerted by several mechanisms, either through classical HIF-1 regulation [13] or a novel oxygen response element (ORE) [14], where transcription is regulated by the interplay of three proteins [12]. Environmental conditions thus add yet another mode of COX regulation to complement nuclear-encoded subunit isoforms that are expressed in tissue or a developmental stage-dependent manner [38]. Induction of COX4i1 expression under hypoxia was reported in multiple cell lines, and it was proposed that the COX4 isoform switch represents adaptation to decreased oxygen availability by improving efficiency of mitochondrial respiration [13]. However, huge variation of Cox4i2 expression in mouse tissues in vivo [14,15] suggested that regulation of its content is complicated beyond just oxygen concentration, and the presence of the second isoform of COX4 may imply its specialized function. Indeed, high expression of Cox4i2 in pulmonary artery smooth muscle cells (PASMCs) of the mouse lung was associated with regulation of pulmonary hypoxic vasoconstriction (HPV) [16]. More recently, increased expression of Cox4i2, along with two atypical COX subunits—COX8b and NDUFA4l2 triggered by HIF-2α—was observed in glomus cells of carotid bodies in mice [17]. Such atypical COX complex present in these specialized cells was suggested as a hypothetical sensor involved in acute oxygen sensing [18].

In the present study, we aimed to identify how the switch between COX4i1 and COX4i2 isoforms is able to modulate COX function. In order to achieve this in a well-defined model, we generated knock-in cell lines of these isoforms independently in the background of COX4i1/2 double knock-out in HEK293 cells to obtain cells with exclusive expression of either isoform. Electrophoretic and MS analyses revealed that COX4i1 and COX4i2 knock-in cell lines were able to restore comparable levels of the COX complex with otherwise identical subunit composition, representing an ideal tool for unbiased characterization of isoform effects. Previously, contradicting results regarding the effect of the COX4 isoform on enzyme activity were reported. COX isolated from mouse lung containing COX4i2 had a higher turnover number than liver COX containing COX4i1 [15]; similarly, transient overexpression of COX4i2 in HEK293T cells mildly augmented COX activity and cellular respiration [13]. In contrast, the isoform switch from COX4i2 to COX4i1 in glioma cells was associated with increased COX activity and respiration [20]. In our KI cell lines, no differences between COX4i1 and COX4i2 KI were observed in the case of COX activity, affinity to cytochrome *c* or respiratory capacities, which suggests that previously published functional modulations may have been associated with regulatory mechanisms dependent on COX4 isoforms but emanating from levels of complexity beyond isoform exchange itself. Despite identical COX activities and respiratory capacities in our KI cell lines, analysis of energetic metabolism in intact cells revealed increased OCR/ECAR ratios in COX4i2 KI cell lines, indicative of the preference for mitochondrial ATP production rather than by anaerobic glycolysis. Both modes of energy provision are utilized by the majority of cultured cell lines, and under conditions of non-limiting substrate (glucose) supply and routine energy demand, mitochondrial OXPHOS activity may be downregulated.

COX has been identified as a target for multiple regulatory mechanisms, including posttranslational modifications and allosteric binding of adenine nucleotides enabling regulation based on the ATP/ADP ratio [4,5]. Indeed, the adenine nucleotide binding site on the matrix-facing domain as well as in vivo phosphorylation sites was identified in the COX4 subunit [4]. In this regard, differential responses of COX4 isoforms to such regulations have been previously described. In astrocytes exposed to hypoxia, robust expression of COX4i2 was identified in contrast to neuronal cells, which retain mostly the COX4i1 isoform [19]. In this study, activity measurements of COX solubilized from astrocytes displayed significantly attenuated allosteric inhibition by ATP, suggesting that COX4i2 is less prone to such modulation than COX4i1 [19]. Further, it has been reported that allosteric ATP inhibition is modulated by the phosphorylation status of Ser58 in COX4i1 [9]. Under conditions of increased flux of the TCA cycle, an increased level of carbonate activates the intramitochondrial PKA pathway, which leads to reversible phosphorylation of Ser58 and impairment of ATP binding through charge repulsion, thus enabling robust flux of electrons through the respiratory chain to match TCA cycle activity. In COX4i2, the corresponding phosphorylation site is missing; instead, negatively charged residues are present in the incriminated region. This suggests that COX4i2-harbouring COX resides are permanently in an activated state [9], matching our observation of the increased OCR/ECAR ratio. Furthermore, this finding corresponds to an increased NAD^+^/NADH ratio and decreased mitochondrial ROS production measured under the normoxic condition in COX4i2 KI cells in comparison with COX4i1 KI.

To enable functional alteration of COX under conditions of limited oxygen availability or to possess the ability to function as an oxygen sensor, COX4 isoforms should confer distinctive oxygen affinity to the enzyme. To the best of our knowledge, this aspect of the COX4 isoform switch has not been previously addressed before in sufficient detail. We analyzed the COX oxygen kinetics by high-resolution respirometry [31]. This method yields the p_50_ parameter, the “apparent K_m_”, a measure of the sensitivity of cellular respiration to oxygen availability assayed in permeabilised cells with intact mitochondrial membranes, where the COX function is integrated in the respiratory chain. Our results consistently showed that oxygen affinity of the COX4i2-containing enzyme was significantly decreased compared to the COX4i1 counterpart. In our opinion, this represents the major finding of this study. The measured values of p_50_ in parental HEK293 cells match previously reported values in isolated mitochondria and small cells [35]. It should also be noted that the two-fold p_50_ increase in COX4i2 versus COX4i1 KI can have significant in vivo implications. For example, similar alteration of COX oxygen affinity can be observed in cells with pathological mutations in the SURF1 protein, which lead to pathologic COX deficiency and is clinically associated with Leigh’s syndrome [29,30]. The mechanism of how COX4 isoform exchange modulates kinetics of oxygen binding is unclear, but up to three dioxygen diffusion areas with open channels relying on rapid and reversible conformational changes were proposed to operate simultaneously to sustain remarkably high affinity of COX to oxygen [39]. Thus, any perturbation in the hydrophobic core may affect oxygen diffusion to the binuclear site within the COX1 subunit. This may be conferred by the transmembrane region of COX4 that runs parallel to the transmembrane helix 12 of COX1, but all available structural data were obtained with the COX4i1-containing enzyme, preventing evaluation of structural differences between isoforms. However, the COX4 subunit capability to modulate enzyme oxygen handling seems to be conserved in eukaryotes, as the expression of the isoform pair of the yeast COX4 homolog is also regulated by oxygen [10]. COX isozymes containing the normoxic isoform COXVa (analogous to COX4i1) or hypoxia-induced COXVb (analogous to COX4i2) were analyzed by infrared spectroscopy in the CO-bound state (CO-IR) as a sensitive way to assess the effects of COXV isoforms on the binuclear reaction center environment [40]. Absorption properties of COXVa-containing COX were similar to those of bovine enzyme and indicated the presence of multiple conformers capable of rapid interconversions. In contrast, narrower CO-IR band of the COXVb isozyme suggested a less dynamic enzyme with only one conformer [40]. If oxygen diffusion to the binuclear center is indeed dependent on dynamic conformational changes, it would mean that the COXVa presence would facilitate it, while rigid conformation of the COXVb-containing enzyme may impose a diffusion barrier. If we speculate that an analogous situation occurs with mammalian COX4 isoforms, this could translate as an increase in p_50_. This issue should, however, be resolved by direct structural comparison of mammalian COX isozymes. In fact, our cell models with the exclusive expression of individual COX4 isoforms may in future prove useful for such studies.

The role of COX4i2 in HPV was associated with mitochondrial membrane hyperpolarization inducing increased ROS production at upstream respiratory complexes. Mitochondrial ROS serve as signaling molecules leading to inhibition of ion channels at the plasma membrane of PASMCs leading to its depolarization and Ca^2+^ influx-triggering contraction [16]. Alternatively, according to our findings in HEK293 KI cell lines, an increase in hypoxic ROS production at complex III that is critical for HPV [41] could be caused by premature inhibition of COX4i2-containing COX due to its decreased oxygen affinity, while sufficient oxygen concentration would be available at ROS-producing sites which are generally more sensitive to oxygen availability than COX [42]. Indeed, in the COX4 knock-out cell line mimicking enzyme inhibition, we observed robustly increased ROS production. Interestingly, p_50_ in COX4i2 KI cells was not modulated by mutations of either of three cysteine residues, which arguably represent the most significant structural difference between COX4 isoforms. In contrast, increased hypoxic ROS production in COX4i2-overexpressing CMT cells was attenuated by cysteine replacement [16]. This suggests that additional modification of COX4i2 may be necessary for affecting proper function as oxygen sensor. Indeed, in glomus cells of carotid bodies, HIF-2α induced expression of COX4i2, but also two other non-canonical COX subunits, namely, COX8b and NDUFA4l2 [17,18]. The latter harbors an extra cysteine residue compared to its ubiquitous isoform, which might hypothetically form a disulphide link with COX4i2. In our study, the immunoprecipitation analysis of HEK293 KI cell lines (Figure 1F) showed COX 4i1 and 4i2 isozymes with otherwise identical isoform composition of other COX subunits. Thus, we can demonstrate that a change in oxygen affinity is solely associated with COX4 isoforms and does not involve modulation via cysteine residues. On the other hand, modulation of hypoxic ROS production may be a more complex phenomenon involving a switch in yet other subunits, present only in physiologically specialized cells that assemble the dedicated oxygen-sensing enzyme.

Overall, our study extends current knowledge on the regulatory role of COX4 isoforms and shows that besides affecting the ability of COX to respond to ATP levels [9,19], the COX4i2 subunit also facilitates a decrease in cytochrome *c* oxidase affinity to oxygen. These findings thus further characterize unique properties of the COX4i2 isoform, in accordance with the view that the COX4i2 subunit represents an important effector in regulation of COX activity and respiratory chain energetic function under hypoxic conditions. According to previously published information and novel findings of the present study, the COX4 isoform exchange regulated by oxygen availability has consequences for cellular respiration as well as mitochondrial ROS production. Activity of the energy status-regulated COX4i1 isozyme is less sensitive to hypoxia; therefore, the enzyme can operate at extremely low oxygen concentrations, limiting non-respiratory oxygen consumption. In contrast, the hypoxia-sensitive COX4i2 isozyme becomes prematurely inhibited under oxygen-limiting conditions, leading to a reduction of upstream components of the respiratory chain, promoting increased ROS production (Figure 7). Further studies are necessary to dissect the interplay of oxygen utilization, regulation of ROS production, and putative redox modification of COX4i2, as well as the role of other non-canonical COX subunits that may be involved in this remarkable physiological phenomenon of oxygen sensing.

## Figures and Tables

**Figure 1 cells-09-00443-f001:**
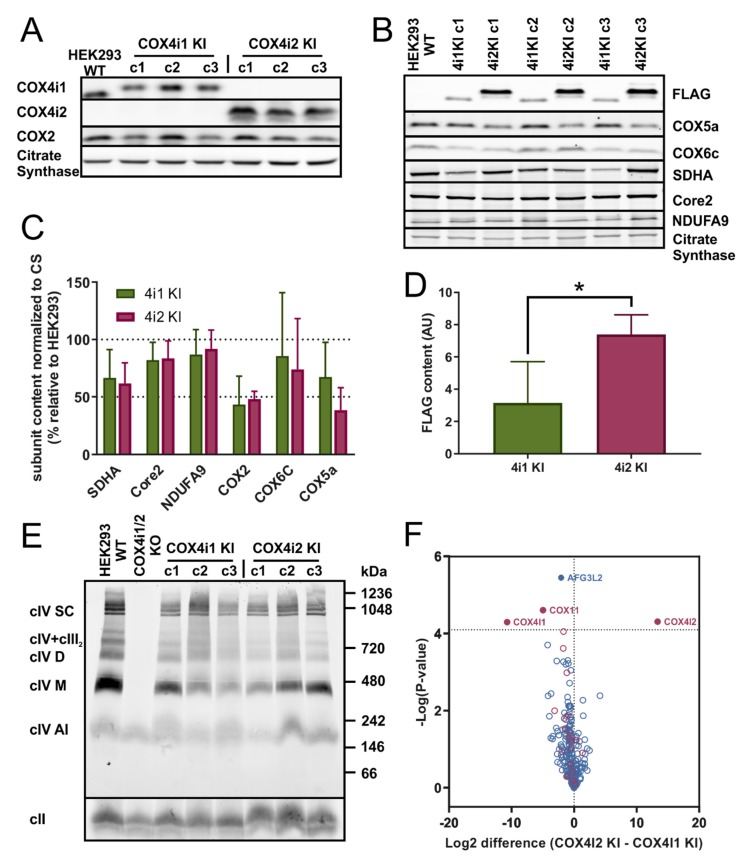
(**A**) Western blot analysis of COX4 isoforms in knock-in cell lines. Whole cell lysates (30 µg of protein) of control HEK293 and three clones of each COX4i1 KI and COX4i2 KI were subjected to SDS-PAGE and Western blotting. COX4i1 and COX4i2 isoforms were detected with specific antibodies (see details in Methods), along with the COX2 subunit. Citrate synthase was used as a loading control. (**B**) Western blot analysis of COX subunits and OXPHOS proteins. Whole cell lysates (30 µg of protein) of control HEK293 and three clones of each COX4i1 KI and COX4i2 KI were subjected to SDS-PAGE and Western blotting. Representative image of one of three sets of samples of the HEK293 control and three clones of each KI cells are shown. (**C**) Quantity of OXPHOS proteins. Antibody signals from WB images were quantified densitometricaly using Image Lab software (Bio Rad). Data are plotted as the mean ± S.D. of COX4i1 and COX4i2 KI values expressed in percent relative to control HEK293 cells (*n* = 9, three independent samples of three clones of each KI cell line). (**D**) Quantity of FLAG-tagged COX4 isoforms. Relative content of FLAG-tagged COX4i1 and COX4i2 in respective KI cell lines determined by densitometry quantification of the FLAG antibody signal. Data are plotted as the mean ± S.D. of COX4i1 and COX4i2 KI cells (*n* = 9, three independent samples of three clones of each KI cell line). (**E**) BN-PAGE analysis of native forms of COX. Isolated mitochondria from HEK293, COX4i1/2 KO, and KI cells were solubilized with digitonin (6 g/g protein); sample aliquots of 30 µg protein were separated on 5%–16% native gel. WB analysis with the COX1 antibody revealed multiple forms of the COX complex as indicated on the left of the image: COX1-containing assembly intermediate (cIV AI), monomer (cIV M), dimer (cIV D), assemblies with complex III (cIV + cIII), and respiratory supercomplexes (cIV SC). Molecular weight marker migration is indicated on the right. Complex II (cII) detected by the SDHA antibody was used as a loading control. * *p* > 0.05. (**F**) Mass spectrometry analysis of immunoprecipitated COX. Volcano plot depicting differential contents of Mitocarta-annotated proteins immunoprecipitated with the anti-FLAG antibody (blue circles) from 4i1 or 4i2 KI cells. Red circles mark COX subunits and assembly factors. Relative quantities (4i2 KI/4i1 KI) plotted on the *X*-axis are expressed in log2 scale. Significantly different hits are shown above the probability cut off (dashed horizontal line) with protein names annotated.

**Figure 2 cells-09-00443-f002:**
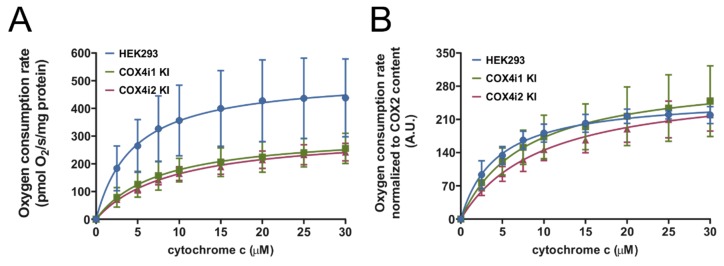
COX activities normalized to the protein concentration (**A**) or COX2 subunit content (**B**). COX activities in Tween-20 extracts of isolated mitochondria measured as oxygen consumption using Oxygraph-2k (Oroboros) at increasing concentrations of cytochrome *c* (0–30 µM) are plotted as the mean ± S.D. value of three (HEK293, blue) and nine (three replicates of three clones each) COX4i1 (green) and COX4i2 KI (magenta) cells each. Hyperbolic fit of the Michaelis-Menten function is shown as full lines in respective colors.

**Figure 3 cells-09-00443-f003:**
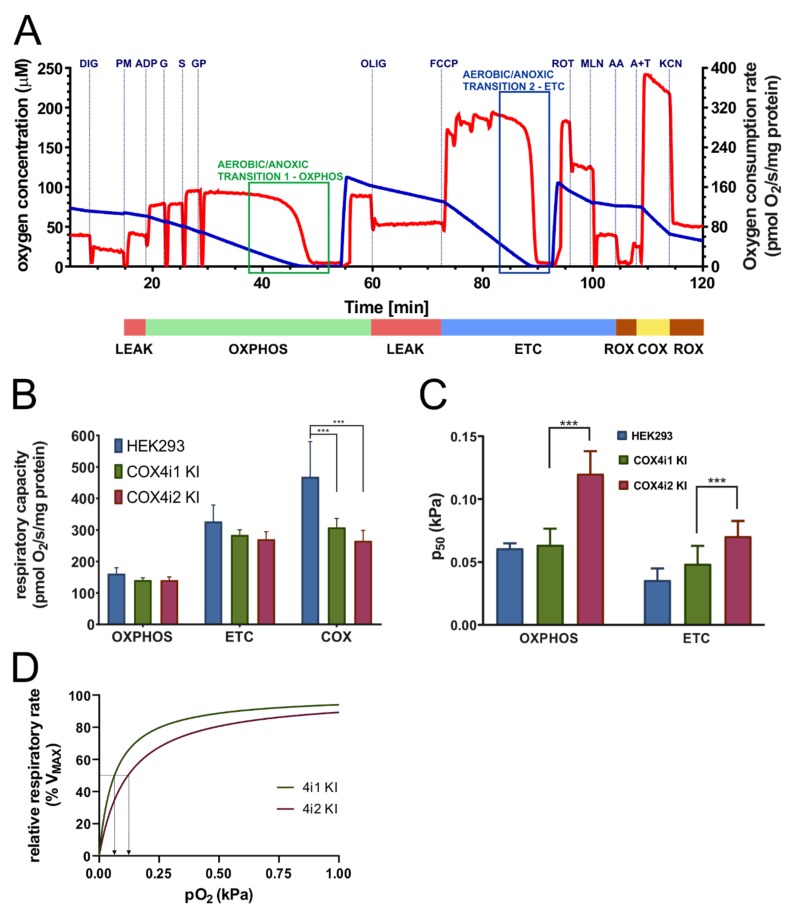
Mitochondrial respiration in permeabilised cells. (**A**) Representative trace of respirometric measurement. Experimental trace recorded by Oxygraph-2k (Oroboros) with 0.3 mg protein/mL of HEK293 cells from acquisition software DatLab5 showing the actual O_2_ concentration (blue, left Y axis in µM) and rate of oxygen consumption (red, right Y axis in pmol O_2_/s/mg protein). Additions of substrates and inhibitors are marked by vertical dashed lines and abbreviations above the trace (digitonin 0.05 g/g protein—DIG, pyruvate 10 mM + malate 2 mM—PM, ADP 1 mM—ADP, glutamate 10 mM—G, succinate 10 mM – S, glycerol-3 phosphate 10 mM—GP, oligomycin 0.5 µM—OLIG, FCCP 100—300 nM—FCCP, rotenone 0.5 µM—ROT, malonate 10 mM—MLN, antimycin A 0.25 µM—AA, ascorbate 2 mM + TMPD 1 mM—A+T, and KCN 0.5 mM—KCN). The diagram below is time-aligned with the experimental trace and denotes the actual respiratory state (driven by proton leak—LEAK, red; coupled to ATP synthesis—OXPHOS, green; uncoupled by protonophore FCCP—ETC, blue; residual oxygen consumption after inhibitor addition—ROX, brown; respiration with artificial substrates ascorbate and TMPD—COX, yellow). (**B**) Respiratory capacities in OXPHOS, ETC, and COX states are plotted as the mean ± S.D. value of seven (HEK293, blue) and twelve (four replicates of three clones each) COX4i1 (green) and COX4i2 KI (magenta) cells. *** *p* < 0.001. (**C**) Oxygen affinity expressed as p_50_—partial pressure of oxygen at half-maximal respiration (kPa) in OXPHOS and ETC states—is plotted as the mean ± S.D. value of seven (HEK293, blue) and twelve (four replicates of three clones each) COX4i1 (green) and COX4i2 KI (magenta) cells. *** *p* < 0.001. (**D**) Simulated hyperbolic response of relative respiratory rates (as % of maximal rates) to oxygen partial pressure (0–1 kPa) was calculated using the Michaelis-Menten function, feeding mean values of experimentally analyzed p50 values for COX4i1 (green) and COX4i2 (magenta) KI cells. Arrows projecting from respective traces towards the *X*-axis mark p_50_ values of COX4i1 and COX4i2 KI.

**Figure 4 cells-09-00443-f004:**
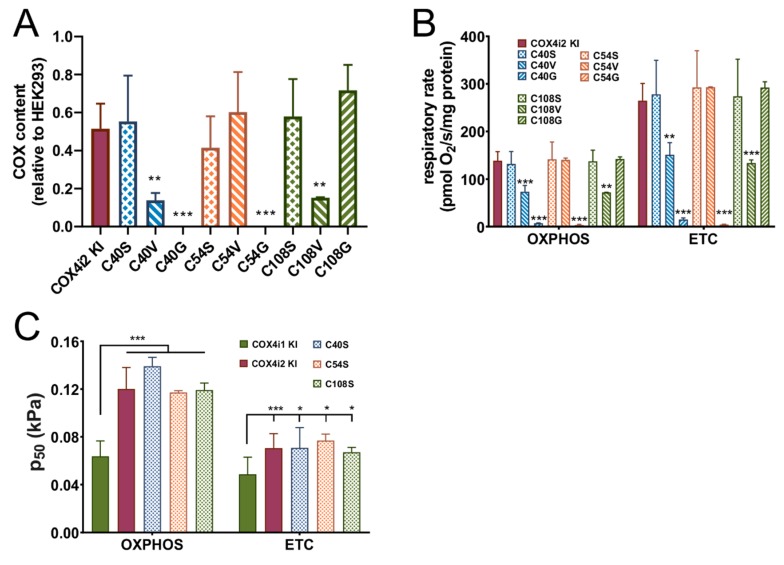
The COX content and respiration of COX4i2 cysteine substitution mutant KI cell lines. (**A**) The COX content calculated as the COX2 subunit quantified in the WB immunodetection experiment, normalized to citrate synthase, and expressed in % relative to the COX2 content in parental HEK293 cells, is plotted as the mean ± S.D. value of two measurements in each COX4i2 cysteine-substitution mutant KI cell line as indicated below the graph compared to wild-type COX4i2 KI (magenta, *n* = 9). ** *p* < 0.01, and *** *p* < 0.001 mark significant differences between individual mutant cysteine KIs and wild-type COX4i2 KI. Respiratory rates (**B**) and p_50_ (**C**) in OXPHOS and ETC states are plotted as the mean ± S.D. value of two measurements in each COX4i2 cysteine-substitution mutant KI cell line as indicated below the graph compared to wild-type COX4i2 KI (magenta, *n* = 12). * *p* < 0.05, ** *p* < 0.02, *** *p* < 0.001 mark significant differences between individual mutant cysteine KIs and wild-type COX4i2 KI in (**B**) or between COX4i1 KI and wild-type or mutant variants of COX4i2 KI cells (**C**).

**Figure 5 cells-09-00443-f005:**
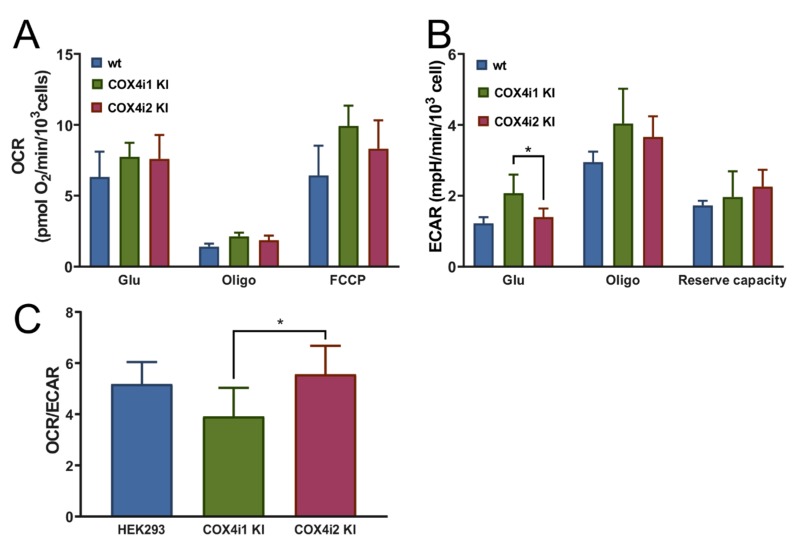
Energy metabolism in intact KI cells. (**A**) Cellular oxygen consumption rate (OCR) and (**B**) extracellular acidification rate (ECAR) were recorded by an XFe Bioenergetics Analyzer (Seahorse) with endogenous substrates (basal) and after subsequent additions of 10 mM glucose (Glu), 1 µM oligomycin (Oligo), 1 µM FCCP (FCCP) in HEK293 cells (blue, *n* = 5), COX4i1 KI (green, *n* = 7, at least two replicates of each of three clones), and COX4i2 KI cells (magenta, *n* = 8, at least two replicates of each of three clones), plotted as mean values ± S.D. Glycolytic reserve capacity displayed in B was calculated as the difference between the ECAR values in Oligo and Glu states. (**C**) OCR/ECAR ratio (mean ± S.D.) was calculated from recorded rates in the coupled state in the presence of 10 mM glucose (Glu). * *p* < 0.05 marks a significant difference between COX4i1 and COX4i2 KI cells.

**Figure 6 cells-09-00443-f006:**
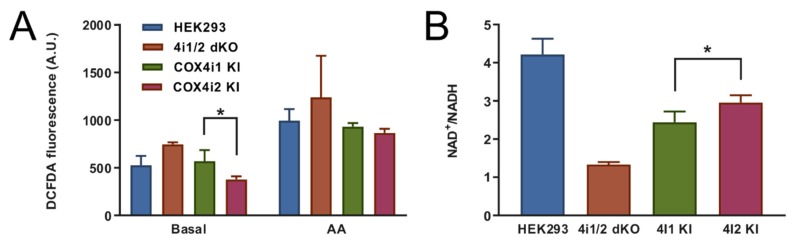
ROS production and redox status in intact KI cells. (**A**) Mitochondrial ROS production was monitored as oxidation of the fluorescent H_2_-DCFDA probe in intact HEK293 (blue, *n* = 3), COX4i1/2 KO (orange, *n* = 3), COX4i1 KI (green, *n* = 9, three replicates of three clones each), and COX4i2 KI cells (magenta, *n* = 9, three replicates of three clones each). Data are plotted the mean ± S.D. (**B**) The NAD^+^/NADH ratio was calculated from parallel measurements of NAD+ and NADH contents in intact cells using NAD/NADH-Glo™ Assay (Promega). Data are plotted as the mean ± S.D. of HEK293 (blue, *n* = 3), COX4i1/2 KO (orange, *n* = 3), COX4i1 KI (green, *n* = 9, three replicates of three clones each), and COX4i2 KI cells (magenta, *n* = 9, three replicates of three clones each). * *p* < 0.05 marks a significant difference between COX4i1 and COX4i2 KI cells.

**Figure 7 cells-09-00443-f007:**
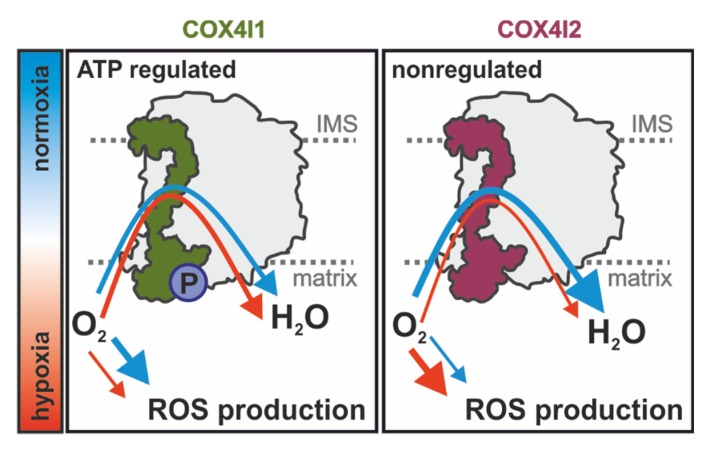
Model of COX functional alterations mediated by COX4 isoform exchange: Rates of oxygen consumption of COX4i1- (**left**) or COX4i2-containing COX (**right**) are depicted by variable arrow thickness in normoxia (blue arrow) and hypoxia (red arrow). P, serine58 phosphorylation site.

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
