# Peer review of "Cytochrome *c* Oxidase Subunit 4 Isoform Exchange Results in Modulation of Oxygen Affinity"

_cells, 2020, doi:10.3390/cells9020443_

Round 1
Reviewer 1 Report
The conclusions of the paper are fully supported by the experimental results.
This is a carefully prepared study on the function of COX subunit IV isoforms. The applied methods of knockin/out of isoform genes are up to date. I fully support the publication of this paper in Cells.
Author Response
We thank the reviewer for considerate review of our manuscript.
Reviewer 2 Report
This is a well-performed study on the role of the two isoforms of subunit IV in mammalian cytochrome c oxidase. Below, I have listed some improvements that the authors may consider
"c" in the title should be lower case and italics
Introduction: It is said that the transmembrane region of subunit IV has extensive interactions with subunit I "comprising the catalytic site and subunit II. This is not evident from the crystal structure, according to which the transmembrane helix of subunit IV lies far from the catalytic site and very far from subunit II. This should be restated.
Why do the authors switch to oxygen pressures (in kPa) when describing the rate dependence on O2 concentration in Fig. 3D while they use molar concentration in the actual measurements (Fig. 3A)?
As the authors are undoubtedly aware, the observed O2 affinity of COX does not describe the actual O2 binding affinity to the active site, but is a kinetic parameter. In this vein it would be interesting for the reader to learn what ideas the authors might have to explain why the 2nd isoform of subunit IV shows a two-fold lower apparent affinity for O2 - especially considering how far subunit IV is from the active site, and especially considering the structural differences between the two isoforms.
